# Psychoacoustic Principle, Methods, and Problems with Perceived Distance Control in Spatial Audio

**Bosun Xie \*,† and Guangzheng Yu \*,†**

Acoustic Lab, School of Physics and Optoeletronics, South China University of Technology, Guangzhou 510641, China
\* Correspondence: phbsxie@scut.edu.cn (B.X.); scgzyu@scut.edu.cn (G.Y.)
† These authors of Bosun Xie and Guangzheng Yu contributed equally to this work.

**Abstract:** One purpose of spatial audio is to create perceived virtual sources at various spatial positions in terms of direction and distance with respect to the listener. The psychoacoustic principle of spatial auditory perception is essential for creating perceived virtual sources. Currently, the technical means for recreating virtual sources in different directions of various spatial audio techniques are relatively mature. However, perceived distance control in spatial audio remains a challenging task. This article reviews the psychoacoustic principle, methods, and problems with perceived distance control and compares them with the principles and methods of directional localization control in spatial audio, showing that the validation of various methods for perceived distance control depends on the principle and method used for spatial audio. To improve perceived distance control, further research on the detailed psychoacoustic mechanisms of auditory distance perception is required.

**Keywords:** spatial audio; psychoacoustics; perceived distance control

## 1. Introduction

Spatial audio or spatial sound aims to simulate (or record), transmit (or store), and, finally, reproduce the spatial information of sound and then to recreate the desired spatial auditory events or perceptions [1]. Spatial audio is conventionally applicable to cinema and domestic sound reproduction. Recently, with the development of computer and information processing techniques, spatial audio has been applied to many fields, such as virtual and augmented reality, communication, and internet audio. Spatial audio techniques are also increasingly serving as tools for scientific research on human hearing and engineering design. Various spatial audio techniques and systems based on different physical and auditory principles have been developed for different purposes.

An ideal spatial audio reproduction should be able to accurately reconstruct the target sound field or binaural pressures to reproduce all spatial auditory cues and, thus, to recreate target spatial auditory events or perceptions. However, this is often infeasible due to the physical restrictions of practical spatial audio techniques. Practical spatial audio techniques are usually only able to create parts of spatial auditory cues. Taking advantage of the redundancy of information provided by different spatial auditory cues, these techniques can be used to recreate desired spatial auditory events or perceptions in spatial audio to some extent. Therefore, the psychoacoustic principles of spatial auditory perception must be understood to create various desired spatial auditory events and perceptions in spatial audio.

One important target spatial auditory event in spatial audio reproduction is the perceived virtual source at various spatial positions in terms of direction and distance with respect to the listener [1,2]. Currently, the technical means for recreating virtual sources in different directions of the various spatial audio techniques are relatively mature. In contrast, the technical means for perceived distance control in spatial audio are often limited, depending on the physical and psychoacoustic principles used in spatial audio.

Multiple acoustic cues contribute to auditory distance perception in real environments [3–6]. Auditory distance perception is a consequence of the comprehensive processing of the information provided by these cues in the auditory system (including high-level systems). However, the interactions and redundancies among various cues are still not completely clear. The dependence of these cues on frequency, source direction and distance, and acoustic environment (reflections) complicates the situation [6]. These problems and complexities have led to the detailed mechanism of distance perception remaining unclear in comparison with that of directional localization. Accordingly, perceived distance control in spatial audio remains a challenging task.

This article reviews the psychoacoustic principles, methods, and problems of perceived distance control and compares perceived distance control with directional localization control in spatial audio. The psychoacoustic principles of auditory distance perception are outlined in Section 2; the principles, methods, and problems of perceived distance control in spatial audio reproduction based on different principles are reviewed in Sections 3–6, respectively. The conclusion is presented in Section 7.

## 2. Psychoacoustic Principles of Auditory Distance Perception

### 2.1. Cues for Auditory Distance Perception

Although human auditory distance estimation ability is generally inferior to that of visual distance estimation, preliminary but biased auditory distance estimation can still be formed [3]. Experiments demonstrated that the human auditory system tends to significantly underestimate distances to sound sources with a physical distance farther than a rough average of 1.6 m, and typically overestimates distances to nearby sound sources with a physical distance of less than a rough average of 1.6 m. This finding suggests that the perceived source distance is not always identical to the physical one. Zahorik examined experimental data from a variety of studies and found that the relationship between the perceived distance $r_I$ and the physical distance $r_S$ can be well-approximated with a compressive power function by using a linear fit method [4]:

$$r_I = \kappa r_S^{\delta},\tag{1}$$

where $\kappa$ is a constant whose average is slightly greater than one (average of approximately 1.32); and $\delta$ is a power-law exponent whose value is influenced by various factors, such as experimental conditions and subjects, so this value varies in a wide range with a rough average of 0.4. The just notice difference (JND) of the percentage variation in distance also varies in different experiments and depends on multiple factors.

Auditory distance perception, which was thoroughly reviewed by Zahorik et al. [5] and Kolarik et al. [6], is the consequence of a complex and comprehensive process based on multiple cues. The known cues are outlined as follows:

(1) Level or loudness cue

In a free field, the sound pressure generated by a point source with constant power is inversely proportional to the distance between the sound source and the receiver (the $1/r_S$ law); that is, the sound pressure level (SPL) decreases by $-6$ dB for each doubling of the source distance. Subjective loudness is closely related to the SPL and considered an effective cue in distance perception. As the source distance increases, the variation in SPL with the source distance decreases. From the $1/r_S$ law, we can derive that the percentage variation in the source distance is related to the variation in the sound level $\Delta L$ in dB as:

$$\frac{\Delta r_S}{r_S} \times 100\% = \frac{1}{\exp(\Delta L/20) - 1} \times 100\%.\tag{2}$$

Therefore, the level or loudness cue provides information on the percentage variation in distance. If a JND of SPL of 1 dB is assumed, the threshold of percentage distance discrimination on the basis of the level cue is about 20%. The experimental results of the JND of percentage variation in distance based on level cue vary from 5% to 25% [6].

The variation in the SPL with source distance derives from the $1/r_S$ law with a rate of less than $-6$ dB for each doubling of a source distance in a reflective room, depending on the acoustic characteristics of the room. The sound pressure and loudness at a listener's position are also closely related to source properties, such as source power and directivity. Prior knowledge of sound sources or stimuli also influences distance estimation performance when loudness-based cues are used. In general, loudness is regarded as a relative distance cue unless the listener is highly familiar with the pressure level of the sound source.

(2) Spectral cue

The high-frequency attenuation caused by air absorption is another cue in auditory distance perception. For a distant sound source, air absorption acts as a low-pass filter and modifies the spectra of the sound pressures at the receiving position. A change in the spectrum may cause perceivable timbre variation. This effect is important only for an extremely far sound source (farther than 15 m) and negligible in an ordinary-sized room [2]. However, this effect is valid for reflections in a room because the accumulative propagation distance for a sound is large after being reflected repeatedly. Moreover, prior knowledge on the sound source may influence the distance estimation performance when a high-frequency spectral cue is used. In general, spectral cues caused by high-frequency attenuation provide weak information for relative distance perception.

(3) Curve wavefront and near-field HRTF cues

In a free field, the sound field radiated by a point source is a spherical wave with a curved wavefront at proximal distance to the source. The scattering and diffraction of the anatomical structures, such as the head and pinnae, to the spherical wave result in distance-dependent binaural pressure even if the effect of the $1/r_S$ law has been compensated. This physical course can be described by near-field head-related transfer functions (HRTFs), which are defined as the normalized pressure at two ears caused by a point source [7]:

$$H_\alpha(r_S, \Omega_s, f) = \frac{P_\alpha(r_S, \theta_S, \phi_S, f)}{P_0(f)}, \tag{3}$$

where $P_\alpha$ denotes the pressure at either ear; $\alpha = $ L or R denotes left or right ear, respectively; and $P_0$ is the pressure at the position of the head center in the absence of the head. Generally, HRTFs vary with frequency $f$, source distance $r_S$, and direction $(\theta_s, \phi_S)$ with respect to the head center. For a source distance farther than 1.0 m, HRTFs are asymptotically independent of the source distance and called far-field HRTFs. HRTFs also vary from individual to individual due to the individual differences in the anatomical structures and the dimensions of the head and pinnae.

Figure 1 illustrates the left and right HRTF magnitudes of a human subject for two horizontal azimuths and three source distances [8]. Figure 1a,b indicate two horizontal source azimuths of $\theta_s = 0°$ and $90°$, which denote the front and left directions, respectively. The variations in HRTF magnitudes with source distance are obvious at a lateral direction of $\theta_s = 90°$.

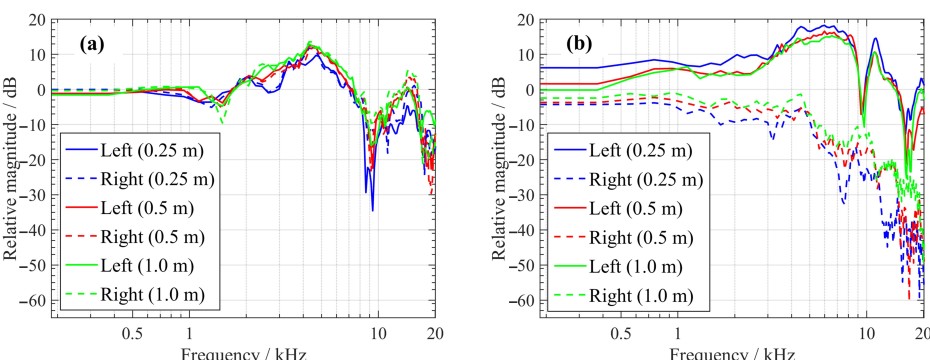

**Figure 1.** The left and right HRTFs magnitudes of a human subject for two horizontal azimuths and three source distances. (**a**) $\theta_s = 0°$; (**b**) $\theta_s = 90°$ .

The auditory distance perception cues encoded in near-field HRTFs are termed near-field HRTF cues, which are valid at proximal source distances of less than 1.0 m (to head center), especially less than 0.5 m.

Near-field HRTFs involve two distance-related physical effects. The first effect is the distance-related interaural level difference (ILD). Outside the median plane, when a source approaches the head, the scattering and shadow effects of the head cause an increase in the pressure at the ipsilateral ear and reduction in the pressure at the contralateral ear. Such a variation in binaural pressures results in a slight variation in interaural time difference (ITD), but a large variation in ILD that increases with the reduction in source distance. As an example, Figure 2 illustrates the ILD versus frequency for a horizontal frontal source at $\theta_s = 0°$, horizontal lateral source at $\theta_s = 90°$, and three different source distances $r_S$ of 0.25, 0.5, and 1.0 m. The ILDs are evaluated from the HRTFs of the same individual as in Figure 1. For the lateral source, the magnitude of ILD increases obviously with the reduction in source distance. For a source distance $r_S$ of 0.25 m and at low frequency of 250 Hz, the ILD is about 10 dB. The distance-dependent ILD, especially at mid and low frequencies below 3 kHz, provides absolute and strong distance perception information for lateral source at proximal distances of less than 1.0 m [9–11]. In addition to ILD, the distance-dependent head scattering and shadow also change the spectra of binaural pressure and change the perceived timbre, which may also provide information for distance perception.

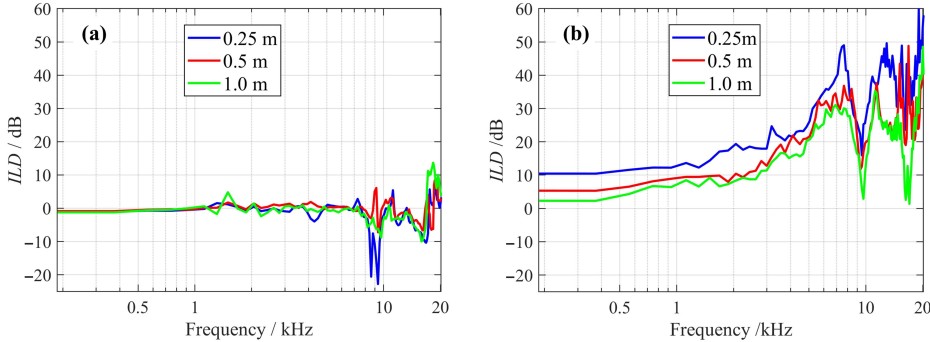

**Figure 2.** ILD versus frequency for a (**a**) horizontal frontal source at $\theta_s = 0°$, (**b**) lateral source at $\theta_s = 90°$ and three different source distances $r_S$ of 0.25, 0.5, and 1.0 m.

Another physical effect involved in near-field HRTFs is the parallax effect [12,13]. When a source in the horizontal plane approaches the head, the source azimuth relative to the (ipsilateral) ear changes, even if source azimuth relative to the head center is unaltered. The change in source azimuth relative to the pinnae changes the effect of the pinnae on incident sounds and, therefore, changes the spectrum of a high-frequency HRTF. The parallax effect may provide absolute but weak distance perception information for a proximal source, especially for a frontal source where the ILD cue is unavailable.

(4) Direct to reverberation energy ratio

In a room with diffused reverberation sound field, the direct-to-reverberant energy ratio (DRR) can be calculated as [14]:

$$\text{DRR} = \frac{D_S \Sigma_S}{16\pi r_S^2} \frac{\alpha_{abs}}{1 - \alpha_{abs}} , \qquad (4)$$

where $r_S$ is the distance between the sound source and the receiver position, $D_S$ is the directional factor of the sound source, $\Sigma_S$ is the total absorption area, and $\alpha_{abs}$ is the average absorption coefficient. Therefore, the DRR is inversely proportional to the square of the source distance. Although the completely diffuse sound field is actually very rare, the DRR serves as an absolute and strong auditory distance perception cue (the absolute distance perception means that the listener can report the exact value of perception distance, not the relative far or near judgement) [15]. This cue is valid for various source distances, and

most effective for a source near the reverberation radius at which the DRR is equal to the unit. However, it is still not completely clear how the auditory system uses the distance information contained in the DRR cue. The possible candidates include the information provided by timbre coloration in the reverberation, the reverberation tails on transients, and interaural coherence in reverberation, although some authors recently argued that interaural coherence may not be used [16].

Inspired by Kolarik et al. [6], the various auditory distance perception cues and their available conditions are summarized in Table 1. Different cues may be involved. Distance perception is enhanced if multiple cues are available. For example, in a free field, distance perception is more accurate when both HRTF and level cues are available [11]. In a reflective room, distance perception is more accurate if both DRR and level cue are available, although there is some controversy on this issue [15,17–19]. However, the information provided by multiple distance perception cues may be somewhat redundant. Distance perception may be still possible in the absence of some cues. For example, distance perception is still possible in a free field in which the DRR cue is absent. Various cues may contribute to distance perception with different weights. These weights vary with multiple factors, such as frequency, source distance and direction, anechoic or reflective environment, etc. Under certain conditions, some cue may be dominant.

**Table 1.** Summary of various auditory distance perception cues. The question mark ('?') indicates that the existing research on this problem has not been clarified.

| Condition | Distance Cue | | | | |
| --- | --- | --- | --- | --- | --- |
| | Level | Spectra | HRTF | | DRR |
| | | | ILD | Parallax | |
| Free field | Yes | Yes | Yes | Yes | No |
| Reverberation field | Yes | Yes | Yes ? | ? | Yes |
| Absolute or relative distance perception | Relative | Relative | Absolute | Absolute | Absolute |
| Source direction | Frontal and lateral | Frontal and lateral | Lateral | Frontal | Frontal and lateral |
| Source distance | Proximal and distant | >15 m | Proximal | Proximal | Proximal and distant |

In summary, auditory distance perception is biased. It is a comprehensive consequence of multiple cues that depend on multiple factors of the sources and acoustic environment. Auditory distance perception cues in a room especially depend on the reflections and, thus, are different from those in a free field. The multiple-factor dependence of distance perception cues, as well as the interaction, cooperation, and redundancy of these cues, complicate the situation. Some mechanisms of distance perception remain unclear. Further research on this issue is needed. These psychoacoustic features of auditory distance perception create challenges for perceived distance control in spatial audio.

### 2.2. Compared with Directional Localization Cues

In contrast to distance perception, auditory directional localization is unbiased. Human hearing is able to localize the direction of a sound source with adequate accuracy. For example, the minimal audible angle of a horizontal-frontal source is 1–3° [2], although the acuity of directional localization decreases for other source directions.

The cues and mechanisms of auditory directional localization in a free field are relatively clear [2,7]. Directional localization is also the consequence of a comprehensive process based on multiple cues. For a far-field sound source, the interaural phase delay difference (ITDp) at low frequencies below 1.5 kHz and the ILD at high frequencies above 2–3 kHz are lateral localization cues. Spectral cues created by the scattering and diffraction

of anatomical structures (especially by pinnae) at high frequencies above 5–6 kHz and dynamic cues (especially dynamic ITD variation at low frequencies below 1.5 kHz) caused by head turning contribute to front-back and vertical localization. Notably, for a source at proximal distance, ITD varies slightly with source and, thus, is not an effective distance perception cue. However, an ILD below 3 kHz and pinna-related spectra are also distance perception cues.

Directional localization is enhanced if multiple cues are available. However, the information provided by multiple directional localization cues may be somewhat redundant. Directional localization may still be possible in the absence of some localization cues or even with the existence of conflicting localization cues. For example, when either a high-frequency spectral cue or a dynamic cue is eliminated, another cue alone still enables vertical and front-back localization to some extent [20]. ITD dominates lateral localization as long as the wideband stimuli include low frequencies regardless of conflicting ILDs [21]. However, if too many conflicts or losses exist in localization cues, the accuracy and quality of localization likely degrade, split virtual sources are perceived, or localization may become impossible [22–24]. As a comparison, distance perception is also achieved through multiple cues, but the relationships and redundancies of distance perception cues are not as clear as those of directional localization cues.

Summing directional localization with multiple sound sources (loudspeakers) is an important psychoacoustic phenomenon [2]. For appropriate multiple sound source configurations (loudspeakers) and correlated source signals, if the relative level and arrival time of each sound sources satisfy certain conditions, the listener perceives a summing virtual source. Summing a virtual source with relative level differences among the sources (interchannel level differences) is due to the superposed sound waves at two ears creating an appropriate ITD at low frequency, which dominates the lateral localization. A summing virtual source with a relative time difference (interchannel time difference) or both relative time and level differences among the sources are psychoacoustic phenomena, but physical interpretations or models for these phenomena have yet to be developed. The principle of summing directional localization is the basis of stereophonic and multichannel sounds' reproduction [25–29]. In contrast, no psychoacoustic phenomenon of summing distance perception with multiple loudspeakers has been observed.

It is also interesting to compare the influence of reflections on directional localization and distance perception. Reverberation may degrade the performance of directional localization. However, providing that the delay and strength of the reflections satisfy the condition of the precedence effect [30], reflections have a slight influence on the perceived direction of a sound source. This effect is important for spatial sound reproduction and helpful for designing acoustic performance and loudspeaker layout in a reflective listening room. In contrast, reflections or reverberation participate in auditory distance perception, which complicates the task.

In summary, auditory directional localization is unbiased and relatively accurate. It also based on multiple factors. The interaction, cooperation, and redundancy of these cues are relatively clear. Especially under the condition of the precedence effect, the influence of reflection on directional localization can be ignored. These psychoacoustic features of auditory directional localization facilitate perceived direction control in spatial audio.

## 3. General Consideration of Perceived Distance Control in Spatial Audio Reproduction

Existing spatial audio techniques can be classified into the following three types according to their principles and methods [1]:

(1) Sound field-based methods

Sound field-based methods aim to reconstruct an original or target sound field as exactly as possible in an appropriate area. Listeners acquire the target spatial information and corresponding auditory perceptions as they enter the reproduced sound fields. High-order ambisonics and wave field synthesis are two examples of these methods [31].

(2) Sound field approximation and psychoacoustic-based methods

Under certain conditions, two physically different sound fields may produce similar spatial auditory perceptions. Psychoacoustic-based methods aim to recreate target auditory perceptions. The sound field reproduced by this method is physically different from the original or target sound field. It is, at most, a rough approximation of the target sound field. With an appropriate number of channels and loudspeaker configuration, this method uses appropriate psychoacoustic principles to recreate auditory perceptions similar to those of the target sound fields. In other words, these methods aim to deceive our hearing [29]. Examples of this method include amplitude or time panning in stereophonic and multichannel sound [32], and the first order Ambisonic signal mixing with psychoacoustic optimization [27].

(3) Binaural-based methods

Binaural pressures or signals embody the auditory information in any complex sound field. Binaural-based methods aim to exactly duplicate and render binaural signals, then to create the desired auditory perceptions. Examples of binaural-based method include binaural recording and reproduction, and virtual auditory displays [7,33].

The first and second types of methods are generally intended for loudspeaker reproduction, and the third type of method is generally intended for headphone presentation. However, after appropriate signal processing, such as HRTF-based virtual loudspeaker synthesis or transaural processing [33,34], the spatial audio signal for loudspeaker reproduction can be converted to signals for headphone presentation, and vice versa.

As stated, the major distance perception cues are the level cue, spectral cue, HRTF cue, and direct to reverberation energy ratio (reflection cue). An intuitive method to control perceived distance is to recreate all desired distance perception cues and information in spatial audio reproduction. However, this method is often infeasible.

The level and spectral cues can be easily controlled and recreated using various reproduction methods. The former is formed by controlling the overall gain in the signals, and the latter can be implemented by an appropriate low-pass filtering of the signals. However, both level and spectral cues are relative rather than absolute cues of auditory distance perception. The practical effects of controlling these two cues only are limited.

Effective methods for perceived distance control in spatial audio strongly rely on effectively controlling HRTF and reflection cues. However, practical spatial audio reproduction is often able to recreate parts of these two cues to some extent, depending on the spatial audio system and the technique in use. In this situation, the knowledge of the interaction, cooperation, and redundancy of different distance cues is important, by which the perceived distance can be controlled by manipulating the major cue being reproduced. However, such knowledge is not complete due to the multiple-factor dependence and complicated interaction between different distance perception cues, which creates challenges for the perceived distance control in practical spatial audio reproduction. Therefore, perceived distance control is more complicated than perceived directional control in spatial audio.

Reflections in the listening room may also spoil the distance perception information in spatial sound reproduction. Distance perception is closely related to reflections. In loudspeaker-based reproduction, the sound waves received by a listener are the superposition of these direct sounds from loudspeakers and reflections from the listening room. When recreating free-field virtual sources at different distances, reflections in the listening room provide misleading distance information. When recreating a virtual source in a target reflective environment, reflections in the listening room decrease the effectiveness of the control of target reflections. Therefore, reflections in the listening room more influence perceived distance control than directional localization control in spatial audio reproduction. For directional localization, the influence of reflections in the listening room can be ignored providing that the condition of precedence effect is satisfied [30]. Appropriate sound absorption processing may reduce the influence of reflections in a listening room on the perceived distance control in audio reproduction.

## 4. Perceived Distance Control in Spatial Audio with Sound-Field-Based Methods

Physically, a sound field in air can be entirely specified by the sound pressure $p(\mathbf{r}, t)$ as a function of the time $t$ and vector $\mathbf{r}$ of the receiver position. Sound-field-based methods aim to reconstruct the original (target) sound field or pressure function $p(\mathbf{r}, t)$ as exactly as possible in an appropriate receiver region. The target sound field cannot be accurate for several listeners since the position is crucial for spatial cues.

Ambisonics is a typical example of sound-field-based methods, which is based on the principle of the spatial harmonics decomposition and reconstruction of a sound field [31,35,36]. In ambisonics, sound fields are decomposed into a linear combination of directional harmonics with different orders (azimuthal Fourier decomposition for horizontal sound fields and directional spherical harmonics functions decomposition for spatial sound fields), and the loudspeaker signals are derived by matching the reconstructed sound field with each order approximation of the target field in the directional harmonics domain. A sufficiently high-order ambisonics (HOA) is theoretically able to be used to reconstruct the free-field plane wave caused by a far-field source within a local receiver region. This is a significant fact. According to the spatial Fourier theorem, an arbitrary sound field within a source-free region can be linearly decomposed into plane waves incident from various directions. Therefore, a sufficiently high-order Ambisonics is theoretically able to be used to reconstruct an arbitrary sound field within a local receiver region, including the early reflections and the late diffused reverberation sound field of a target room or environment, so as to exactly recreate the spatial perception information related to reflections, including source distance perception information. Moreover, by matching the reconstructed sound field with each order approximation of a target spherical wave, a near-field compensated higher-order Ambisonics (NFC-HOA) is theoretically able reconstruct the curve wavefront caused by a virtual point source within a local receiver region and thereby recreate the distance perception cues of a proximal source in a free field [37]. In practice, the Ambisonics signals, which include directional localization and distance perception information, can be obtained by simulation or recording with circular or spherical microphone arrays.

Wave field synthesis (WFS) is another example of sound-field-based methods [38,39]. According to the Huygens–Fresnel principle and Kirchhoff–Helmholtz boundary integral equation, the frequency-dependent pressure $P(\mathbf{r}, f)$ in an arbitrary, closed, and source-free space is determined by the pressure and its normal derivative on the surface. Accordingly, the sound field within the boundary surface can be theoretically reconstructed by an array of dipole and monopole loudspeakers arranged on the boundary surface. The signals of dipole and monopole loudspeakers are directly proportional to the target pressure and its normal derivative on the boundary surface, respectively. In certain situations, the target sound field can be reconstructed by a simplified array including only one type of loudspeaker (such as a monopole loudspeaker). WFS is theoretically able to reconstruct an arbitrary target sound field within an extended region enclosed by the loudspeaker array. By using a time-reversal technique, WFS is able to create the sound field of a focused virtual source that is located at a proximal distance within the loudspeaker array [40]. Therefore, similar to Ambisonics, WFS is theoretically able to be use to accurately create various directional localization and distance perception cues, including distance perception cues related to direct and environmental-reflected sound. For practical horizontal WFS (the 2.5-dimensional WFS), the mismatched radiation characteristics of loudspeakers cause the distance dependence of the reconstructed sound field to somewhat deviate from that of the target sound field [39].

The sound-field-based methods function on the basis of sound field spatial sampling and reconstruction. Limited by the Shannon–Nyquist spatial sampling theorem, practical sound-field-based methods are only able to reconstruct the target sound field up to a certain frequency. According to this theorem, an $L$-order spatial Ambisonics requires $(L + 1)^2$ loudspeakers at least, and is able to reconstruct a target sound field within a region of

radius $r_0$ and up to a frequency limit of $f_{max}$ [41]. The upper frequency is approximately related to the radius and order of ambisonics according to the following equation:

$$f_{max} = \frac{cL}{2\pi r_0},$$ (5)

where $c$ is the sound speed of 340 m/s. Therefore, within a region with a size of the mean head radius $r_0$ of 0.0875 m, the preceding five-order ambisonics are able to accurately reconstruct the target sound field up to the frequencies of 0.62, 1.25, 1.87, 2.47, and 3.1 kHz, respectively. Accordingly, 4, 9, 16, 25, and 36 loudspeakers at least are required for reproduction, respectively. If an accurate reconstruction of a target sound field is desired within the same region up to a frequency limit $f_{max}$ of 20 kHz, at least $L = 32$-order spatial Ambisonics with $(L + 1)^2 = 1089$ loudspeakers are required. Such high-order Ambisonics is infeasible in practice.

Similarly, for WFS, according to spatial sampling theorem, the interval between two adjacent loudspeakers should not exceed half of the wavelength (the worst case scenario). Considering the size of a practical loudspeaker, the interval between two adjacent loudspeakers is between 0.1 and 0.3 m, at least. Accordingly, the upper frequency limit for accurate sound field reconstruction is between 1.7 and 0.57 kHz. Reconstructing a target sound field up to an upper frequency limit of 20 kHz in WFS is also infeasible. Therefore, practical sound field-based methods are usually only able to reconstruct the target sound field up to an upper frequency limit of 2–3 kHz, or up to 4 kHz, at most, thus being only able to recreate parts of directional localization and distance perception information. This is a major problem with the sound-field-based methods. In this situation, various psychoacoustic principles and means must be incorporated into the sound-field-based methods to create the desired spatial auditory events.

Taking advantage of the redundancy of various directional localization information and the dominant role of low-frequency ITD in lateral localization, sound-field-based methods can create perceived virtual sources in different directions, provided that it is able to accurately reconstruct the target sound field up to 1.5–2.0 kHz. For example, theoretical analysis indicated that third-order Ambisonics are able to accurately recreate ITD and its dynamic variation with head turning up to at least 1.5 kHz for a central listening position and, thus, provide appropriate information for directional localization [28,42]. Virtual source localization experiments also proved that spatial Ambisonics can recreate a virtual source in three-dimensional directions (including vertically). The localization performance improves as the order increases. Third- or forth-order Ambisonics usually exhibits satisfactory directional localization performance [28,43–45]. Based on a similar psychoacoustic principle, in practice, WFS is able to recreate a perceived virtual source in different directions [46].

Similarly, the psychoacoustics and redundancy of various distance perception cues may be beneficial to perceived distance control in the sound-field-based methods. For example, using a practical loudspeaker array for reproduction, Favrot and Buchholz demonstrated that a fourth-order NFC-HOA with appropriate regularization on a near-field source distance coding filter was able to create different distance perceptions in lateral directions to some extent even when the level cue was removed [47]. Because ILD below 3 kHz is a distance perception cue for proximal lateral source, a fourth- or fifth-order Ambisonics is able to provide a distance-dependent ILD cue below 2–3 kHz. Incorporating the level and high-frequency spectral cues, a third- to fifth-order Ambisonics is expected to be able to recreate the major distance perception cues in a free field. An enhanced scheme to improve the accuracy of ILD control in Ambisonics was proposed [48]. In reflective environments, the direct to reverberation energy ratio is a distance perception cue. Some experiments demonstrated that the first-order binaural Ambisonics is sufficient to recreate auditory distance perception in a reflective environment [49]. Based on a similar psychoacoustic principle, in practice, WFS is able to recreate free-field perceived virtual sources at different distances [50], although the experiment, which was based on a dynamic

binaural reproduction experiment, indicated that in practice, WFS may cause some artifacts in recreating focused source.

As in all cases of loudspeaker-based reproduction, reflections in the listening room may spoil the distance perception information in the sound-field-based methods. However, the field-based methods, including Ambisonics and WFS, are also able to cancel or compensate for listening room reflection [51,52]. The compensation is theoretically valid below the upper frequency limit imposed by Shannon–Nyquist spatial sampling theorem. Incorporation of listening room reflection compensation may improve the accuracy of target distance information reproduction and enhance the perceived distance control in sound-field-based methods. Further investigations and experiments in this area are needed.

Overall, practical sound-field-based methods are only able to reconstruct the target sound field up to certain frequency limit and the psychoacoustic principle should be incorporated to recreate different auditory distance perceptions. Because the knowledge of the interaction, cooperation, and redundancy of different distance cues is incomplete, further investigations are needed.

### 5. Perceived Distance Control in Sound Field Approximation and Psychoacoustic-Based Methods

These methods are used in many practical spatial sound techniques and systems, such as conventional two-channel stereophonic sound and various multichannel sounds. Because the reproduced sound field by this type of methods is a rough approximation of the target sound field, at most, these methods are unable to accurately create all spatial perception cues. These methods rely completely on the psychoacoustic principle to recreate various spatial auditory events and perceptions. For example, the psychoacoustic principles of summing directional localization with two or more loudspeakers are usually used with these methods to create perceived virtual sources in various directions [25–29]. By feeding decorrelated signals to multiple loudspeakers, it is also possible to create a sensation of envelopment in reproduction [53]. Based on these principles, various microphone and signal panning/mixing techniques for stereophonic and multichannel sounds have been designed [32,54,55].

Because no psychoacoustic principle of summing distance perception with multiple loudspeakers is available, the methods for perceived distance control in stereophonic and multichannel sound are limited. In addition to controlling the level and spectral cues in reproduction, a common practice to create different distance sensations in stereophonic and multichannel sound is to control the proportion of reflections/reverberations in the reproduced signals using different signal-mixing or microphone-recording techniques. Adding reflection and reverberation make the perceived virtual source farther away from the listener, although this method can not make the perceived virtual source closer to the listener. However, stereophonic and multichannel sound reproductions are unable to accurately recreate reflection/reverberation sound fields, and thereby may be unable to recreate all reflection-related distance information. Moreover, reflections in a listening room, which are difficult to be compensated for in stereophonic and multichannel sound reproduction, may influence the reproduction of reflection-related target distance information. Therefore, the effect of the existing methods of perceived distance control in stereophonic and multichannel is limited. As the number of channels increases, some sound-field-based methods (such as the HFC-HOA method) are able to recreate the distance perception information and compensate for the listening room reflections in multichannel sound reproduction. This is an example of the mixed use of spatial audio techniques based on different principles.

Overall, recreating various auditory distance perceptions in stereophonic and multichannel sound reproduction remains a challenging task; more psychoacoustic research on this issue is needed.

### 6. Perceived Distance Control in Binaural-Based Methods

Binaural-based methods create binaural signals in a target sound field by binaural synthesis processing or artificial head recording, and reproduce the resultant signals

by headphone [7]. The former is termed the virtual auditory display (VAD) or virtual auditory environment (VAE) technique and the latter is termed the binaural recording and reproduction technique. In a static VAD, the binaural signals of a free-field virtual source are synthesized by filtering the input stimulus with a pair of HRTFs at the target source positions or, equally, by convolving with a pair of head-related impulse responses (HRIRs), which are the time-domain counterparts of HRTFs and related to HRTFs by inverse Fourier transform. Moreover, according to the temporary position and orientation of a listener's head detected by a head tracker, a dynamic VAD updates the signal processing of binaural synthesis in real time and provides dynamic information for spatial auditory perception. The binaural signals in a reflective environment can also be synthesized by convoluting the input stimulus with a pair of binaural room impulse responses (BRIRs), which are the acoustic impulse responses from a source to two ears in a reflective environment.

By careful equalization, the non-ideal transmission in the signal and reproduction chain (such as the headphone to ear canal transfer functions), the binaural-based methods are theoretically able to reconstruct binaural pressures in an arbitrary sound field and, therefore, to accurately recreate various spatial information, including directional localization and distance perception information. Moreover, it is free from disturbances caused by listening room reflections by using headphone presentation. Therefore, the schemes for perceived distance control in binaural-based methods, especially in VAD, are relatively mature. VAD also allows various perceived cues (including distance cues) in binaural signals to be manipulated individually; therefore, it is an effective experimental tool in psychoacoustic studies related to spatial auditory perception. Many psychoacoustic experiments related to auditory distance perception have been conducted on the basis of VAD [56]. However, there are still some problems with the perceived distance control in binaural-based methods, especially in VAD.

(1) Problems with dynamic cues in binaural-based methods

Inappropriate binaural reproduction by headphone is prone to creating in-head localization (intracranial lateralization), a psychoacoustic phenomenon in which an auditory event is perceived inside the head. In-head localization often occurs for free-field virtual sources with a frontal target direction in static VAD. The possible reasons for in-head localization include errors in the reconstructed binaural pressures at the eardrums, the absence of reflections in the simulation of the free-field virtual source, and omission of dynamic cue in a static VAD.

Externalization of an auditory event is essential to perceived distance control with the binaural-based methods. Although dynamic cues caused by head turning may not contribute obviously to distance perception for a real sound source, it is important for the complete externalization of an auditory event in VAD, especially the externalization of a frontal free-field virtual source [57]. Therefore, in addition to careful equalization of non-ideal transmissions in the signal and reproduction chain, a dynamic VAD is essential for perceived distance control, at least in the cases of scientific research with VADs. Otherwise, the in-head-localization effect may spoil the experimental results. However, some previous experiments were based on static VADs.

(2) Problems with near-field HRTFs

Near-field HRTFs are essential for recreating virtual sources at different proximal distances in a VAD. HRTFs depend on individuals. Although psychoacoustic experiments indicated that individualized features in near-field HRTFs have little influence on the distance perception of a proximal virtual source [58], they may influence the directional localization in a VAD. Therefore, individualized near-field HRTFs are needed for accurate binaural synthesis in VADs, at least for most scientific research purposes. Measurement is a direct and accurate method to obtain near-field HRTFs. However, near-field HRTF measurement is relatively difficult [8]. The distance-dependent characteristics of near-field HRTFs require time-consuming measurements for various source directions and distances. In addition, a special point sound source in the near field is needed.

Brungart et al. measured the near-field HRTFs of a KEMAR artificial head in sparse directions in their pioneer work on auditory distance perception [9]. However, only a few other groups measured the near-field HRTFs of artificial heads and set up a database in recent years [59,60]. Using a special spherical dodecahedron sound source and the first-generation measurement system established by our group, Yu et al. measured the near-field HRTFs for KEMAR with DB 60/61 small pinnae [61,62]. The sound source and the scene of the first generation of near-field HRTF measurement system are shown in Figure 3. The resultant database included HRIRs with 10 source distances of 0.20, 0.25, 0.30, 0.40, 0.50, 0.60, 0.70, 0.80, 0.90, and 1.00 m, and 493 directions at each source distance. Using the second-generation system of our group (the fast measurement system), Yu et al. further measured the near-field HRTFs of 56 human subjects with seven source distances and 685 directions at each distance, establishing the near-field HRTF database of human subjects [8]. The layout of the second generation of near-field HRTF measurement system with many loudspeakers can be found in Figure 4. The above two near-field HRTF databases are for scientific research purposes, and the KEMAR database is available from the SOFA website since 27 March 2020 (https://www.sofaconventions.org/). Notably, in some work, the source of the near-field HRTF measurement cannot be considered a point source, which may cause error in the resultant data. Therefore, the resultant database "should be considered as a valuable set for auralization purposes rather than as a basis for sensitive listening experiments", as stated by the authors of the database [60]. Moreover, due to the enormous amounts of data produced by measurements, all the near-field HRTF data should be carefully checked prior to use for scientific purposes to determine if artifacts are presented in some data.

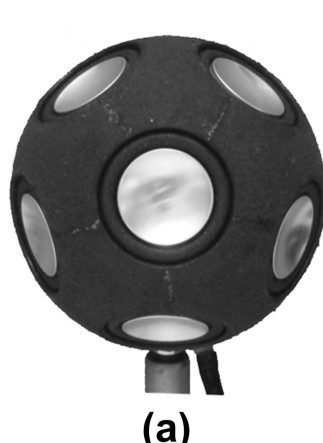 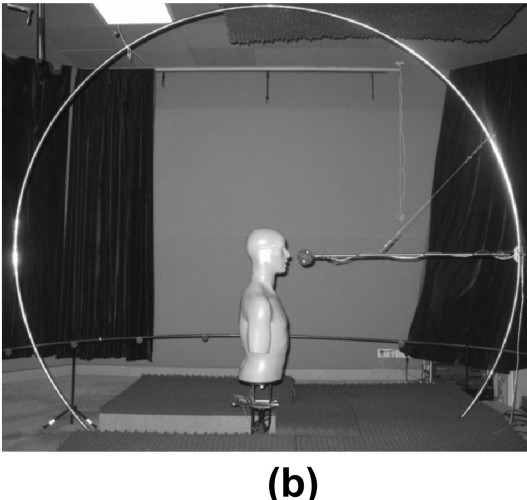

**(a)** **(b)**

**Figure 3.** Photos of the sound source and the scene of the first generation of near-field HRTF measurements system: (**a**) spherical dodecahedron sound source; (**b**) scene of measurements.

Calculation is another method to obtain near-field HRTFs. A spherical head is a simplified model for near-field HRTF calculation [63]. Improved models consist of a spherical head and spherical torso (snowman model) [64], or spherical head, spherical torso, and cylindrical neck [65]. However, these models neglect the influences of head shape and pinnae and, thus, are only valid below 3 kHz. To improve the accuracy of calculation, the geometrical surfaces of the head and pinnae are acquired by a scanning device and the HRTFs are calculated by some numerical methods, such as the boundary element method (BEM). The BEM is able to calculate the near-field HRTFs up to a frequency of 20 kHz [66,67].

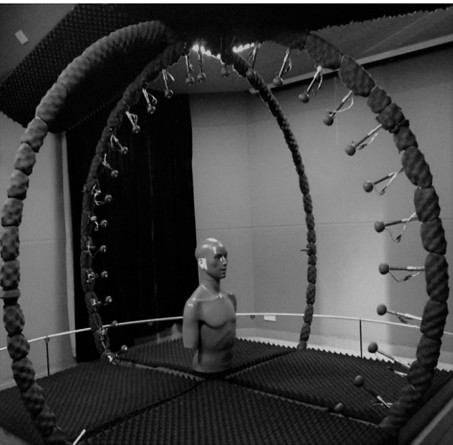

**Figure 4.** Layout of the second generation of generation of near-field HRTF measurement system with many loudspeakers positioned at different elevation angles in a non-anechoic room.

Near-field HRTFs can also be estimated from measured far-field HRTFs. One method involves decomposing HRTFs in arbitrary directions and distances using the spherical Bessel and spherical harmonic functions [68,69]. The weights of decomposition are evaluated from a set of measured far-field HRTFs. A continuous direction and distance representation of near-field HRTFs can be obtained by substituting the resultant weights into the decomposition. This method yields near-field HRTFs with appropriate accuracy but usually requires a far-field HRTF dataset in full spatial directions with sufficient directional resolution (usually more than 1000 uniform measured directions are needed to derive the near-field HRTFs up to 20 kHz). Measuring far-field HRTFs with such uniform directional resolution is also difficult.

Alternatively, Kan et al. suggested a simplified method for estimating near-field HRTFs by multiplying the distance variation function (DVF) from the far-field HRTFs [70]. The DVF is calculated from a spherical head model. This method yields a rough approximation of near-field HRTFs. The resultant data are appropriate for consumer applications of VADs. Researchers should be careful when applying these data for scientific purposes.

(3) Problems with the simplifying signal processing in VADs

With delicate binaural synthesis processing, a VAD is theoretically able to accurately duplicate the binaural pressures of a target sound field. However, limited by computational resources, the binaural synthesis processing in a practical VAD is usually simplified.

Binaural or virtual Ambisonic method is able to simplify the synthesis of free-field virtual sources at different proximal distances in a dynamic VAD [71]. This method includes two stages. In the first stage, the NFC-HOA signals for loudspeaker reproduction are created. In the second stage, the loudspeaker signals are converted into binaural signals for headphone presentation via HRTF-based binaural synthesis. In other words, the NFC-HOA signals are reproduced by virtual loudspeakers. This also provides an example of the mixed use of spatial audio techniques based on different principles.

One advantage of the dynamic binaural Ambisonics method is that only the Ambisonics signals, not HRTF-based filters, are needed to be updated in signal processing after head turning, which simplifies signal processing and avoids the artifacts caused by updating the HRTF-based filters in the conventional scheme. Another advantage of the dynamic binaural Ambisonics method is that only the far-field HRTFs in the virtual loudspeaker directions are needed in binaural synthesis, avoiding the difficulty of acquiring near-field HRTFs. The binaural Ambisonics method is mathematically equivalent to estimating near-field HRTFs from known far-field HRTFs by spherical harmonic decomposition to an appropriate order [7]. Our recent experiment indicated that combined with the level cue, fifth-order dynamic binaural ambisonics is able to recreate the auditory perception of a virtual source in various directions and distances closer than 1.0 m [72].

The scheme of synthesizing environmental reflections should also be simplified. The length of a pair of complete BRIRs is in the order of the reverberation time of the target room, which varies from some hundreds of millisecond for an ordinary room to about two seconds in a concert hall. Updating and convoluting with such BRIRs in a dynamic VAD are difficult. Parameterized room acoustics and BRIRs models are often used in practical VADs, in which the binaural signals caused by direct sounds and a few early reflections are accurately synthesized; the later diffused reverberation is approximately synthesized by appropriate artificial reverberation schemes [73]. The distance perception can also be controlled by adjust the DRR in the binaural signals [74,75]. This simplified processing may also lose some spatial information, including distance perception information related to reflections.

Overall, binaural-based methods are theoretically able to create various spatial auditory cues. To create appropriate distance perception information, dynamic VADs and near-field HRTFs are needed. Simplification of binaural synthesis processing in practical VADs may lose some spatial information, including distance perception information. Further research on the influence of simplifying binaural synthesis on the distance perception in VADs, especially in dynamic VADs, is required.

## 7. Conclusions

Distance perception cues include level, spectral, curved wavefront, HRTF, and direct to reverberation energy ratio (reflections) cues. The level and spectral cues are relative, and the spectral cues are weak. The curve wavefront and HRTF cues are absolute cues for proximal source distances of less than 1.0 m in the lateral direction in a free field. The direct to reverberation energy ratio is an absolute cue and the most effective cue at a distance near the reverberation radius. Various distance perception cues depend on frequency, source distance and direction, and the acoustic environment in a complex manner. The interactions and redundancies of various cues are also complicated. These complexities have been under-researched in terms of the mechanism of distance perception in comparison with directional localization.

To control the perceived auditory distance, ideal spatial audio reproduction is required to accurately create all major distance perception cues. However, practical spatial audio techniques are usually only able to recreate parts of the distance perception cues. Creating level and spectral cues in reproduction is relatively easy; accurately creating curve wavefront and HRTF cues, as well as the reflection cue, is relatively complicated and difficult, depending on the spatial audio principles and methods used. Moreover, reflections in the listening room may influence the accurate control of reflection-related distance perception cues.

The three basic categories of principles and methods for spatial audio reproduction include the sound-field-based methods, sound field approximation and psychoacoustic-based methods, and binaural-based methods. Sound-field-based methods, such as Ambisonics and WFS, are theoretically able to reconstruct various target sound fields and thereby to recreate various spatial auditory perception cues, including distance perception cues. However, limited by the Shannon–Nyquist spatial sampling theorem, in practice, sound-field-based methods are only able to reconstruct a target sound field up to certain frequency limit, usually in the order of 2–3 kHz, or 4 kHz, at most. Psychoacoustic principles should be incorporated to recreate different distance perceptions. Moreover, sound-field-based methods may be able to compensate for the reflections in a listening room.

Sound field approximation and psychoacoustic-based methods, including conventional stereophonic and multichannel sound, are unable to accurately create all spatial perception cues. They rely completely on the psychoacoustic principle to recreate various spatial auditory events and perceptions, including directional localization and distance perception. In contrast to the control of directional localization, the means for perceived distance control in stereophonic and various multichannel sounds are limited. They usually

rely on the control of the direct to reverberation energy ratio in the reproduced signals. However, the reflections in a listening room may hinder such control.

Binaural-based methods are theoretically able to create various spatial auditory cues. Schemes of perceived distance control in binaural-based methods are relatively mature. As a result, a VAD often serves as the experimental tool in psychoacoustic research related to spatial auditory perception. However, the in-head-localization in the headphone presentation must be eliminated and accordingly a dynamic VAD is needed. Near-field HRTFs are required to synthesize free-field virtual sources at proximal distances. Accurate acquisition of near-field HRTFs is relatively complicated; only a few near-field HRTF databases are available. The various approximate near-field HRTFs datasets are mainly valid for consumer applications. Simplifying signal processing in VADs may result in the loss of some distance perception information.

Overall, in comparison with the relatively mature means for perceived direction control, perceived distance control in spatial audio remains a challenging task. Further investigations on the psychoacoustics of auditory distance perception, especially the interaction, cooperation, and redundancy of different distance cues, are needed.

**Author Contributions:** Conceptualization, B.X. and G.Y.; methodology, B.X. and G.Y.; writing—original draft preparation, B.X.; writing—review and editing, B.X. and G.Y.; visualization, G.Y.; supervision, B.X.; project administration, B.X.; funding acquisition, B.X. All authors have read and agreed to the published version of the manuscript.

**Funding:** This research was funded by the National Natural Science Foundation of China (No. 12174118).

**Institutional Review Board Statement:** Not applicable.

**Informed Consent Statement:** Not applicable.

**Data Availability Statement:** Not applicable.

**Conflicts of Interest:** The authors declare no conflict of interest.

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
