# Peer review of "Psychoacoustic Principle, Methods, and Problems with Perceived Distance Control in Spatial Audio"

_applsci, doi:10.3390/app112311242_

Round 1
Reviewer 1 Report
Figure 1 can also be deleted
Furthermore, in reading the paper I did not understand what the authors contribute to the field of the psychoacoustic principle and perceived distance control in spatial audio.
As written, it looks like a review paper, where the contribution of the authors is not clear.
These are topics known to researchers dealing with psychoacoustic principles.
Furthermore, when explaining the acoustic measurements, it is necessary to describe the measurement locations and how the acoustic measurements were performed, and the instrumentation in use.
So in my opinion the paper should be divided into a synthetic introduction of the state of the art and then insert the contributions of the authors.
Reviewer 2 Report
The article reviews current knowledge on spatial audio and recreation of virtual sound sources. The focus is in reproduction of direction and distance information in the perceived sound field at the listeners' position. The psycho-acoustic principles and methods of measurement and modelling are discussed. Certain problems considering distance cues and directional cues are pointed out and some solutions are suggested.
Some detailed comments:
Introduction should have references to support the various statements presented.
Figure 2 shows three distances while the text says two distances. It is not clear, either, what are the HRTFs in back and right directions.
line 116: 3 kHz is not low frequency, should perhaps say in a different way. Low frequencies refer typically below 200 Hz.
line 132: completely diffuse sound field is actually very rare.
Table 1. What does mean absolute distance perception in this context? This could be clarified in the text.
lines 169-170: ITD is defined interaural time difference in introduction. Text should be consistent.
line 225: references to the described methods should be presented
lines 229-235: references to the described methods should be presented
lines 237-240: references to the described methods should be presented
line 285: target sound field cannot be accurate for several listeners since the position is crucial for spatial cues.
line 483: reference to measurement method should be given
Reviewer 3 Report
Authors showed summary of the perceived virtual sources in the spatial audio. Authors showed distance control for spatial audio. There are no English grammar mistakes. The article looks fine. However, authors do not provide some references in the introduction even though authors mentioned some information.
Therefore, the manuscript need to be revised according to the comments as below.
1. Data availability section is missing.
2. In Ref. [7], there is no city information.
3. This article is review paper so authors must change this to review instead of article.
4. Authors mentioned that "These authors contributed equally to this work.". However, there is no mark in the author lists.
5. In Chapter 6, authors provide many information about the perceived distance control. Therefore, authors had better provide some Tables to be easily understood for readers.
Round 2
Reviewer 1 Report
accept
Author Response
Thank you very much for your comments and suggestions on our manuscript. The comments and suggestions are valuable and very helpful for revising and improving our manuscript. And thank you for your final comments and suggestions of “accept” for our manuscript. We will try our best to improve the quality of this article before officially publication.